# High-Fructose Diet Alters Intestinal Microbial Profile and Correlates with Early Tumorigenesis in a Mouse Model of Barrett’s Esophagus

**DOI:** 10.3390/microorganisms9122432

**Published:** 2021-11-25

**Authors:** Andrea Proaño-Vasco, Theresa Baumeister, Amira Metwaly, Sandra Reitmeier, Karin Kleigrewe, Chen Meng, Michael Gigl, Thomas Engleitner, Rupert Öllinger, Roland Rad, Katja Steiger, Akanksha Anand, Julia Strangmann, Robert Thimme, Roland M. Schmid, Timothy C. Wang, Michael Quante

**Affiliations:** 1Department of Internal Medicine II, Klinikum Rechts der Isar, Technical University Munich (TUM), 81675 München, Germany; andrea.proano-vasco@uniklinik-freiburg.de (A.P.-V.); theresa.baumeister@uniklinik-freiburg.de (T.B.); akanksha.anand@tum.de (A.A.); rolandm.schmid@tum.de (R.M.S.); 2Department of Internal Medicine II, Universitätsklinik Freiburg, Universität Freiburg, 79106 Freiburg, Germany; julia.strangmann@uniklinik-freiburg.de (J.S.); robert.thimme@uniklinik-freiburg.de (R.T.); 3Chair of Nutrition and Immunology, Technical University Munich (TUM), 85354 München, Germany; amira.metwaly@tum.de (A.M.); sandra.reitmeier@tum.de (S.R.); 4ZIEL—Institute for Food & Health, Core Facility Mikrobiom, Technical University of Munich (TUM), 85354 München, Germany; 5Bavarian Center for Biomolecular Mass Spectrometry (BayBioMS), Technical University of Munich (TUM), 85354 München, Germany; karin.kleigrewe@tum.de (K.K.); chen.meng@tum.de (C.M.); michael.gigl@tum.de (M.G.); 6Institute of Molecular Oncology and Functional Genomics, Klinikum Rechts der Isar, Technical University Munich (TUM), 81675 München, Germany; thomas.engleitner@tum.de (T.E.); rupert.oellinger@tum.de (R.Ö.); roland.rad@tum.de (R.R.); 7Department of Pathology, Klinikum Rechts der Isar, Technical University Munich (TUM), 81675 München, Germany; katja.steiger@tum.de; 8Department of Medicine, Columbia University Irving Medical Center, New York, NY 10032, USA; tcw21@cumc.columbia.edu

**Keywords:** Barrett’s esophagus, esophageal adenocarcinoma, IL-1B-mouse model, Western diet, fructose, microbiota, metabolism

## Abstract

Esophageal adenocarcinoma (EAC) is mostly prevalent in industrialized countries and has been associated with obesity, commonly linked with a diet rich in fat and refined sugars containing high fructose concentrations. In meta-organisms, dietary components are digested and metabolized by the host and its gut microbiota. Fructose has been shown to induce proliferation and cell growth in pancreas and colon cancer cell lines and also alter the gut microbiota. In a previous study with the L2-IL-1B mouse model, we showed that a high-fat diet (HFD) accelerated EAC progression from its precursor lesion Barrett’s esophagus (BE) through changes in the gut microbiota. Aiming to investigate whether a high-fructose diet (HFrD) also alters the gut microbiota and favors EAC carcinogenesis, we assessed the effects of HFrD on the phenotype and intestinal microbial communities of L2-IL1B mice. Results showed a moderate acceleration in histologic disease progression, a mild effect on the systemic inflammatory response, metabolic changes in the host, and a shift in the composition, metabolism, and functionality of intestinal microbial communities. We conclude that HFrD alters the overall balance of the gut microbiota and induces an acceleration in EAC progression in a less pronounced manner than HFD.

## 1. Introduction

The increase in many industrialized countries of obesity, metabolic syndrome, and associated comorbidities suggests that food choices have an impact on the pathogenesis of diverse diseases, particularly cancer [1,2]. The intake of energy-dense, processed foods rich in refined sugars, salt, and saturated fat is common in industrialized nations, where obesity-associated cancer types occur more frequently. It has been suggested that these food components lead to modifications of the gut microbiome, metabolic changes in the host, and chronic inflammation, which favors carcinogenesis [3].

Esophageal adenocarcinoma (EAC) is one of many cancer types associated with obesity. Over the past decades, the incidence of EAC in most Western industrialized countries has increased at a rate exceeding that of any other cancer, with a 5-year survival rate that remains poor [3,4,5,6]. The risk factors for EAC include white race, older age, male gender, tobacco use, high waist circumference, obesity, and gastroesophageal reflux disease (GERD), but the greatest risk factor is Barrett’s esophagus (BE) [3,4]. BE is an acquired premalignant condition of the distal esophagus characterized by the replacement of normal squamous epithelium by columnar epithelium above the gastroesophageal junction (GEJ), with varying degrees of intestinal metaplasia [4,6,7]. BE appears to be the result of severe esophageal mucosa injury, followed by aberrant epithelial repair [6].

Dietary modifications can lead to microbial alterations in the gut, with effects on stem cell biology and susceptibility to gastrointestinal neoplasia. Considering that environmental or dietary factors prevalent in obese patients influence the risk of BE progression, we recently investigated the effects of a high-fat diet (HFD) on the progression of BE in the L2-IL1B mouse model [8]. The L2-IL1B mouse model overexpresses IL-1B in the esophagus and phenocopies human BE [3]. We showed that HFD-fed L2-IL1B mice developed esophageal dysplasia and tumors more rapidly than mice fed a control diet (CD) and linked this neoplastic acceleration to shifts in the gut microbiota with an increased esophageal infiltration of immune cells [8]. As the mice did not develop obesity but maintained a constant weight, possibly due to increased metabolism related to the systemic inflammatory state, we concluded that the alterations to the microbiome and not the high body weight per se led to neoplastic acceleration. In addition to high fat intake, a westernized diet comprises a high intake of refined sugars, which can lead to increased inflammation [9] and progression of chronic diseases. Processed foods and beverages contain concentrated amounts of refined sugars and, thus, high concentrations of fructose. When consumed in fruits and vegetables, the intake of fructose is accompanied by vitamins, micronutrients, and fiber, whereas the intake of refined fructose has been linked with the metabolic syndrome [10]. Previous in vitro experiments have shown the potential of fructose to induce proliferation and cell growth [11] in pancreatic cancer cell lines [10], immortalized pancreatic ductual cells [12], and human colon adenocarcinoma cells [13]. A high-fructose diet (HFrD) was shown to alter gut microbiota and enhance the inflammatory changes in a dextran sulfate sodium (DSS)-induced colitis model [14]. However, the potential effect of HFrD on other gastrointestinal diseases such as esophageal adenocarcinoma remained unclear. In regard to the increased consumption of refined fructose, a better understanding of the link between high-fructose intake and disease is of great importance. In general, a part of the ingested food escapes digestion in the small intestine, reaches the colon, and is fermented by gut microbes. Thus, composition and functionality of gastrointestinal microbiota is influenced by the dietary habits of the host, since different microbial strains favor certain environment and nutrients [15]. Based on previous results [8] and considering that fructose alters the intestinal microbiota [15], the aim of the present work was to investigate whether a HFrD accelerates esophageal carcinogenesis in a manner similarly to HFD, as both dietary components favor obesity and potentially cancer if consumed excessively.

## 2. Materials and Methods

### 2.1. Animals

For this study, mice of the BE-mouse model L2-IL1B [4], overexpressing human IL1B under control of the EBV-L2 promoter were kept under specific-pathogen-free (SPF) conditions and fed with control diet or high-fructose diet (CD, HFrD; Ssniff, Spezialdiäten GmbH, Soest, Germany) from 2 to 6 or 9 months of age in randomized cohorts. The 6-month cohort was composed of 21 mice, 11 mice were HFrD fed and 10 mice were CD fed. The 9-month cohort was composed of 20 mice, 10 mice were HFrD fed and 10 mice were CD fed. The animal experimental work of this study was conducted under the approval of the district government of Upper Bavaria, according to the animal experimental permit 55.2-1-54-2532-24-2016. All animal experiments were performed in accordance with German Federal Animal Welfare and Ethical Guidelines of the Klinikum Rechts der Isar, TUM, Munich.

### 2.2. Tissue Preparation and Disease Evaluation 

Macroscopic scoring of the squamocolumnar junction (SCJ) and the esophagus was performed on images of the opened and cleaned stomach of all mice. Histopathologic scoring and estimation of the percentage of goblet-like (GC) cells at the SCJ on HE stained formalin-fixed paraffin embedded (FFPE) tissue sections were performed by an experienced gastroenterologist following previously established criteria [3]. Mucus production was quantified as a percentage of periodic-acid Schiff (PAS)-positive area in the BE region of at least 5 mice per intervention group and age cohort. For further phenotypic characterization, immunohistochemistry (IHC) for Caspase 1, γ-H2AX, and Ki67, as well as in-situ hybridization (ISH) for Lgr5 in the BE region of at least 5 mice per intervention group and age cohort were performed.

### 2.3. Microbiome Analysis

Amplifying the V3/V4 region of microbial 16s rRNA genes through 16S rRNA gene amplicon analysis has been established as a standard approach to investigate the microbial diversity of samples [16] and was used in this work to assess diet-related changes in the gut microbiota. Fecal samples of CD and HFrD mice of different age stages (3–4, 6, and 9 months) were collected along the study and sent to the TUM Core Facility Microbiome for DNA isolation, 2-step-amplification-PCR-based library construction and sequencing. The 16S rRNA gene amplicon libraries were sequenced using the Illumina MiSeq (Ilumina, San Diego, CA, USA). To control for artifacts and ensure reproducibility between runs, two negative controls (a PCR control without template DNA and a DNA extraction control of 600 μL stool stabilizer without stool), as well as a positive control using a mock community (No. D6300, ZymoBIOMICS, Irvine, CA, USA) were included in every batch of 45 samples. Generated raw reads were assigned to their corresponding samples by using the barcode pairs unique to each of the samples (demultiplexing). Sequencing resulted in an average total number of 28,070 ± 9888 demultiplexed reads per sample. An average of 20,276 ± 7202 of raw reads per sample remained after filter trimming and removal of chimeras. The negative controls had 16 and 11 reads, respectively, after quality filtering, indicating no systematic contamination in the dataset. The final dataset comprised 385 zOTUs in 75 samples. Sequencing data were preprocessed using the IMNGS pipeline [17]. Data processing was achieved by trimming five nucleotides on the 5′ end and 3′ end for the R1 and R2 read, respectively (trim score 5), and an expected error rate of 1 followed by chimera removal [18] using the FASTQ mergepair script of USEARCH [19]. Quality filtered reads were merged, deduplicated, and clustered, and a denoised clustering approach was applied to generate zOTUs [20]. Taxonomy was assigned using the RDP classifier version 2.11 and confirmed using the SILVA database [21]. EzBioCloud database [22] was used for precise identification of zOTU sequences of interest. The demultiplexed reads for all 16S amplicon sequencing datasets included in this manuscript are publicly available. Raw data of the 16S rRNA gene amplicon analysis have been deposited to the NCBI Sequence Read Archive [http://www.ncbi.nlm.nih.gov/bioproject/769229; accessed on 7 October 2021] under the accession no. PRJNA769229. Software used to analyze the data is either freely or commercially available. The function prediction based on 16S rRNA gene sequencing was performed by using PICRUSt2 [23]. To determine differences between the groups, the R package ALEDx2 was used [24,25,26]. Predicted abundances were centered-log transformed, and significance was determined by using a generalized linear regression. Pairwise significance between visits, within one intervention group (Intervention or placebo), was assigned if the *p*-value <0.05 and effect size >0.4. Centered-log-transformed values of the selected pathways were correlated with zOTUs with a relative abundance >0.1 and a prevalence >10%. zOTUs with a negative or positive correlation in at least one pairwise comparison >0.5 and a significant *p*-value <0.05 were selected to generate the heatmap. The functionality of the pathways found to be significantly different between the intervention groups or to be altered within the groups at different age stages was verified in the database MetaCyc. Additionally, aiming to identify features characterizing the differences [27] between both intervention groups, we performed a LEfSe analysis at genera level. We then looked into the taxonomy browser of NCBI to check the taxonomy of the genera found to be enriched in the intervention groups.

### 2.4. Metabolic Analysis

Untargeted metabolome analyses and targeted analyses for short-chain fatty acids (SCFAs) were performed to assess the metabolic profile in feces and serum of 5 CD mice and 10/11 HFrD mice of each cohort, as well as in the cardia of 5 CD and 5 HFrD mice of each cohort. Untargeted metabolomics were performed using a Nexera UHPLC system (Shimadzu, Nakagyo-ki, Kyoto, Japan) coupled with a Q-TOF mass spectrometer (TripleTOF 6600, AB Sciex, Framingham, MA, USA). Separation of the samples was performed using a HILIC UPLC BEH Amide 2.1 × 100, 1.7 μm analytic column (Waters Corp., Milford, MA, USA), and a reversed-phase Kinetex XB-C18, 2.1 × 100, 1.7 μm analytic column (Phenomenex, Torrance, CA, USA), respectively. Targeted SCFA analysis was performed using a QTRAP 5500 triple quadrupole mass spectrometer (Sciex, Darmstadt, Germany) coupled with an ExionLC AD (Sciex, Darmstadt, Germany) ultrahigh performance liquid chromatography system. According to a method previously described [28], a multiple reaction monitoring (MRM) method was used for the detection and quantification of SCFA.

### 2.5. Immune Cell Analysis via Flow Cytometry

Tissues of 5 CD mice and 10/11 HFrD mice of each cohort were analyzed by flow cytometry for immune cells, using two panels for myeloid and T cell detection. Tissues were processed for flow cytometry analysis, as previously described [29] and stained with the following antibodies: APC-anti-F4/80, APC-e780-anti-cd11b1β, Alexa700-anti-Ly6G, eFluor450-anti- CD45, PE-Ly6C, eFluor450-anti-CD4, APC-CD8a, FITC- anti-CD3, APCe780-NK1.1, and PE-anti-gamma delta TCR (all eBioscience, San Diego, CA, USA), with the addition of 7-AAD to quantify live cells. Flow cytometry data were acquired on a Gallios flow cytometer (Beckman Coulter, Brea, CA, USA) and analyzed using FlowJo™ Software (Becton, Dickinson, and Company, Franklin Lakes, NJ, USA) using compensation and gating templates previously generated (Appendix A). Cell populations are represented as the percentage of cells compared with all CD45+ cells.

### 2.6. Gene Expression Analysis

The gene expression profile of enzymes relevant for fructose metabolism in the liver and colon of 5 CD and 5 HFrD mice of each cohort was analyzed by quantitative polymerase chain reaction (qRT-PCR) and normalized to GAPDH and Cyclophilin A as housekeeping genes. The total gene expression profile of the esophageal tissue from the SCJ of 5 CD- and 5 HFrD-fed mice of the 9-month cohort was assessed by RNA-sequencing.

### 2.7. Statistics

Unless stated otherwise, the statistical analysis of most of the data was performed with GraphPad Prism 8.0.2. for Windows (GraphPad Software, San Diego, CA, USA) by first checking the normal distribution of the data and then using a test accordingly. Normally distributed data were analyzed with either unpaired *t*-tests or 1-way or 2-way analysis of variance (ANOVA). For 1-way ANOVA, Tukey’s multiple comparisons test or Holm–Sidak’s multiple comparisons test was used. For 2-way ANOVA, Sidak’s multiple comparisons test was used. Non-normally distributed data were analyzed with either a Mann–Whitney test or Kruskal–Wallis test. The test used and p-values are noted in each corresponding figure. For the statistical analysis of the gut microbiota, a Kruskal–Wallis rank sum test of all groups and a pairwise Wilcoxon rank sum test between the different groups were performed.

## 3. Results

### 3.1. HFrD Accelerates Dysplasia by Increasing Stem Cell Expansion While Decreasing Mucus Production

The phenotypic effects of a 64%-fructose containing diet (HFrD) compared with control diet (CD) (Appendix A) on the L2-IL1B mouse model were analyzed by evaluating macroscopic and histologic scores of the BE region at the SCJ and by assessing mucus-producing goblet-like cells (GC), inflammation scores, DNA damage, cell proliferation, and stem cell expansion in the BE region at 6 and 9 months. While inflammation and metaplasia scores did not differ between groups (Figure 1A–C), HFrD-fed mice showed higher dysplasia scores (Figure 1D), a lower percentage of goblet-like cells (Figure 1E), and a lower mucus production rate (PAS staining) in the BE region (Figure 1G). However, these differences only reached significance in 6-month-old mice (and not in 9-month-old mice). Expansion of Lgr5-positive progenitor cells in the BE region increased significantly in 6-month-old HFrD compared with CD mice (Figure 1F). Despite higher food intake, no changes in relative and final body weight could be detected between CD and HFrD mice at either the 6 m or 9 m timepoint (Appendix A). We detected a significantly increased weight of the liver in 9-month-old HFrD mice, while the weight of the spleen was significantly increased in 6-month-old HFrD mice compared with CD-fed mice (Appendix A). No difference in macroscopic tumor scores (Appendix A) between the intervention groups was detected. Additionally, HFrD did not cause significant changes in inflammasome activation (Appendix A), DNA damage (Appendix A), or cell proliferation (Appendix A) in the metaplastic tissue at the SCJ compared with CD.

### 3.2. HFrD Decreases Gut Bacterial Richness with Increased Firmicutes and Decreased Akkermansia Abundance

To assess diet-related changes in the gut microbiota, 16S rRNA gene amplicon analysis was performed [16]. In general, HFrD-fed mice showed a lower microbial richness than CD-fed mice. While there were no significant differences in richness between CD-fed L2-IL-1B mice of different ages, an age-dependent decrease in microbial richness was observed in HFrD mice (Figure 2A). Regarding the relative abundance of *Lactobacillus*, there was no significant difference between the intervention groups. However, a significant age-dependent increase in the relative abundance of *Lactobacillus* was observed in both intervention groups (Figure 2B). In general, HFrD mice showed an age-dependent increase in relative abundance of Firmicutes (Figure 2C). The relative abundance of Firmicutes in 6-month-old and 9-month-old HFrD-fed mice was significantly higher than in age-matched CD-fed mice. The relative abundance of Bacteroidia increased significantly in an age-dependent manner in mice in both intervention groups (Figure 2D). Regarding the relative abundance of *Akkermansia*, an age-dependent significant drop was observed in both intervention groups (Figure 2E). The relative abundance of *Akkermansia* in 9-month-old HFrD mice was significantly lower than in 9-month-old CD mice. We also observed an age-dependent increase in the relative abundance of Clostridia in HFrD-fed mice (Figure 2F). A general view of the phyla distribution in the intervention groups over time shows an overall increase in Firmicutes and Bacteroidota (Figure 2G) in both intervention groups and an overall decrease in Verrucomicrobiota in both intervention groups. In contrast, Actinobacteriota was mainly found in samples of HFrD-fed mice. Moreover, while inflammation and dysplasia correlated positively with Bacilli, Lactobacillales, *Lactobacillaceae*, and *Lactobacillus**,* the GC ratio as a marker for differentiation showed a negative correlation with these bacteria. *Akkermansia* and, interestingly, also Clostridia and Lachnospiraceae correlated negatively with dysplasia scores and positively with GC ratio (Figure 2H). The LEfSe analysis performed at genera level showed an enrichment of *Dubosiella*, *Atopopobiaceae*, *Alistipes*, *Erysipelotrichales*, *ASF356*, *Peptostreptococcaceae*, *Marvinbryantia*, and *Erysipelotrichaceae* in HFrD-fed mice. On the contrary, *Bacteroides*, *Oscillospirales*, *Muribaculaceae*, and *Akkermansia* were enriched in CD-fed mice (Figure 2I). For metagenome prediction and functional profiling of the microbial communities present in the gut of IL-1B mice fed with the different diets, a PICRUSt2 analysis of the 16S rRNA gene sequencing data was performed. When looking for diet-related metabolic differences in intestinal microbial communities, we found that pathways associated with the synthesis of nucleic acids (PRPP-PWY, PWY-6125, PWY-7184, PWY-7196, PWY-7200, PWY-7211, PWY-7228), the synthesis of proteins and NAD(P) coenzymes (PRPP-PWY), de novo biosynthesis of folates (FOLSYN-PWY, PWY-6612), and biosynthesis of menaquinones (MK) (PWY-5838, PWY-5897) were significantly upregulated in HFrD-fed mice in comparison with CD-fed mice (Figure 2J). When assessing age-related metabolic differences in the intestinal microbial communities of the IL-1B mice fed with the different diets, we only detected age-related alterations in HFrD-fed mice and not in CD-fed mice. Pathways found to be altered over time in microbial communities of HFrD-fed mice were the TCA pathway and pathways associated with the synthesis of nucleic acids (DENOVOPURINE2-PWY, PWY-6125, PRPP-PWY, PWY-7184, PWY-7187, PWY-7197, PWY-7196, PWY-7200, PWY-7211, PWY-7228), synthesis of methionine (HSERMETANA-PWY), and synthesis and elongation of fatty acids (FASYN-ELONG-PWY). Most of these pathways show higher activity in 3-month-old and 9-month-old mice and lower activity in 6-month-old mice (Figure 2K). Comparing age-related differences in CD, no significantly different pathways were found. Appendix A shows the rarefaction curves for every sample, as well as the graphs resulting from alpha and beta-diversity analysis 

### 3.3. HFrD Alters Colonic and Hepatic Subtrate Utilization, Changes Metabolic Profile of the Host, and Decreases Protective SCFA Concentrations in the Gut and Serum

To explore the metabolic changes caused by the diet, we measured metabolites in feces, serum, and cardia tissue of HFrD- and CD-fed L2-IL1B mice using LC-MS/MS-based metabolomics. When analyzing the untargeted metabolomic data, we focused on the most concise metabolic changes, with emphasis on metabolites enriched only in CD or HFrD samples. Criteria for metabolite identification were peak shape, intensity, retention time of reference compounds, database search, and MS1 and MS2 comparison. HMDB V4.0 [30], MS-DIAL public MS/MS database [31,32], and the in-house database of BayBioMS were used for annotation. Metabolites were considered identified when the retention time and the MS2 spectra matched to the reference. In cases where no reference or database match was available, the MS2 spectra were analyzed with SIRIUS4 (https://bio.informatik.uni-jena.de/sirius/; accessed on 17 June 2021) [33]. In cases where the MS1 and MS2 had several possible annotations due to, for example, different sugar moieties, the name of the overall chemical group was stated.

While a deoxyhexose (MS2) was found in fecal samples of only HFrD-fed mice, a ketodisaccharide (MS1, good peak shape, high intensity, ~60% score in SIRIUS) was found in fecal samples of only CD-fed mice (Figure 3A,B). The intensity of a deoxyhexose was constant in fecal samples of 3- and 6-month-old HFrD-fed mice and decreased significantly in 9-month-old mice (Figure 3A). The intensity of the ketodisaccharide increased significantly with age in 3–9-month-old CD mice (Figure 3B). In serum samples, a deoxyhexose (MS2) was found in only HFrD-fed mice (Figure 3C). While the single hexose (MS2) was found in the cardia tissue of HFrD-fed mice, a trisaccharide (MS2) was found in the cardia tissue of CD-fed mice (Figure 3D,E).

Results of metabolome analysis specifically targeting short-chain fatty acids (SCFAs) in fecal samples from different collection timepoints showed a significantly lower concentration of overall SCFA levels in HFrD-fed mice of all age stages in comparison with age-matched CD-fed mice. The concentration of acetic acid was significantly lower in 3-month-old HFrD-fed mice compared with controls (Figure 4A). The concentration of propionic acid was significantly lower in HFrD mice of all age stages than compared with age-matched controls (Figure 4B). Butyric acid concentrations were significantly lower in 6- and 9-month-old HFrD-fed mice than compared with the corresponding CD-fed mice (Figure 4C). Isobutyric and valeric acid concentrations were significantly lower only in 9-month-old HFrD-fed mice compared with CD-fed mice (Figure 4D,E). Metabolome analysis targeting SCFAs in serum showed a significant decrease in the concentration of butyric acid in 9-month-HFrD-fed mice and a significant drop in the concentration of lactic acid in 6-month-old-HFrD-fed mice compared with controls (Figure 4F,G).

To gain insight into fructose-specific metabolic changes caused by the diet, the expression of enzymes relevant for fructose metabolism in the colon and liver was evaluated in L2-IL-1B mice by qRT-PCR (Appendix A). 6-month-old HFrD mice showed a significantly higher expression of Glucose-6-Phosphatase and Aldolase in the colon (Appendix A). Expression of Triokinase and Fructose Bisphosphatase followed the same trend, yet the changes were not significant. In the colon of 9-month-old mice and the liver of 6-month-old mice, no significant difference in the enzyme expression between the intervention groups was recognized (Appendix A). In the liver of 9-month-old HFrD mice, a significant upregulation of Hexokinase and a significant downregulation of Glucose-6-Phosphatase were observed (Appendix A). Considering the substrates and products of these enzymes, it appears that the fructose-rich diet altered the colonic gluconeogenesis rate and hepatic de novo lipogenesis.

### 3.4. Neutrophil and γδ T Cell Infiltration at the SCJ Tissue of HFrD-Fed Mice Increased with Age

The assessment of immune cell infiltrates in the SCJ tissue showed a significant age-dependent increase in γδ T cells and neutrophils in HFrD-fed mice (Figure 5A,B). In the esophagus, although the amount of neutrophils tended to be lower in HFrD-fed mice than in CD-fed mice of both timepoint cohorts, the effect was only significant in 9-month-old mice (Figure 5C). Appendix A shows representative gating schemes for myeloid and T cells.

### 3.5. HFrD Promotes Changes in Tissue Reconstruction, Immune Network, and Metabolic Pathway

The gene expression profile of the SCJ was assessed by bulk RNA-sequencing. Principal component analysis (PCA) showed a clear separation of the gene expression profiles between groups (Figure 6A). When screening for changes in fructose metabolizing enzyme gene expression, we found that relative gene expression of *Aldolase B*, one of the key enzymes of fructose metabolism, was significantly higher in the tissue of HFrD mice compared with in CD mice (Figure 6B). We performed gene set enrichment analysis using MSigDB Hallmark, KEGG, and Reactome gene sets and selected for the top 10 significantly enriched gene sets (Figure 6C,D). Analysis of the top significantly regulated gene sets led to the finding that tissue reconstruction, intestinal immune network, and metabolic pathways were enriched in HFrD mice (Figure 6C), while in CD mice, pro-oncogenic, gene transcription, and keratinization-associated pathways were enriched (Figure 6D). A clear upregulation of epithelial–mesenchymal transition (EMT), bile acid metabolism, intestinal IgA production, epithelial structure maintenance, and metabolic-disease-associated genes was found in HFrD mice (Figure 6E–H; Appendix A). On the other hand, clear enrichment of genes involved in mTORc, PI3K-Akt-mTOR, and p53 signaling was detected in CD mice (Appendix A).

## 4. Discussion

### 4.1. Diet-Related Alteration of Microbial Profile and Stability Indicates Impaired Gut Health and Correlates with Inflammation and Disease 

Similar to the observations made when assessing the effects of chronic HFD treatment on the acceleration of EAC [8], HFrD-fed L2-IL1B mice showed higher dysplasia scores, a lower percentage of goblet-like cells, and a lower mucus production rate in the BE region. However, these differences only reached significance in mice of the 6-month-old cohort. Goblet cells are highly differentiated specialized cells that mainly produce and secrete protective mucus [34,35,36] and, hence, prevent self-digestion [34]. Additionally, 6-month-old HFrD mice showed increased Lgr5 stem cell expansion in the BE region compared with age-matched controls. Findings showing increased expression of the fructose transporter GLUT5 in undifferentiated cells than in mature cells after fructose treatment [37,38] indicate that undifferentiated cells are a better target for fructose than mature cells. Therefore, it seems that HFrD accelerates dysplasia by reducing differentiation and protective mucus production in BE epithelium, especially at an early timepoint.

A link between an impaired gut micro-ecosystem and disease has been established [39]. Dietary elements not only have an effect on the host’s health, but also on intestinal microbiota [40,41,42]. Results from this study revealed a diet-related alteration of the gut microbiota profile, together with decreasing microbial richness in HFrD mice. Similarly, in HFD-fed L2-IL1B mice, a diet-related alteration of the gut microbiota linked with BE progression was also found [8]. Obesity and Western-style diet have been previously linked with a shift in the gut microbiota, a reduction in Bacteroidetes, and increasing levels of Firmicutes [43]. Fitting into this context, in HFrD-fed L2-IL1B mice, we found no significant reduction in Bacteroidia but increasing levels of Firmicutes. In regard to our results, the age-related increase in Firmicutes in HFrD-fed mice can be explained through the age-dependent increase in Clostridia and the enrichment of Firmicutes genera (*Dubosiella, Erysipelotrichales, ASF356, Peptostreptococcaceae, Marvinbryantia*, and *Erysipelotrichaceae*) in HFrD-fed mice. Clostridia and some members belonging to *Eubacteria* can metabolize primary bile acids to secondary bile acids [44]. The increase in Clostridia might, thus, be related to increasing bile acid metabolization upon HFrD intake, consistent with the gene expression analyses of SCJ tissue from these mice. However, no changes in bile acid levels could be detected in untargeted metabolomic analyses. Within the Clostridia class, there are species shown to have beneficial health properties [45,46]. Considering this, the high relative abundance of Clostridia in HFrD-fed mice together with the fact that Clostridia negatively correlates with dysplasia and positively with GC ratio might explain the less pronounced acceleration of EAC progression in HFrD-fed mice.

*Lactobacilli* have gained attention as potential probiotics [47,48]. Proportions of these bacteria in the colonic microbiota have been described to correlate both positively and negatively with disease and different types of cancer [48,49,50]. In HFrD mice, we showed an age-dependent increase in *Lactobacillus* and observed that the presence of *Lactobacillus* positively correlated with inflammation and dysplasia. Together with decreasing lactic acid levels, these findings suggest that *Lactobacillus* strains promoting disease progression, but not the probiotic, lactic-acid-producing strains, expanded in the gut of HFrD mice. Linked with the decrease in Verrucomicrobiota in both intervention groups, we observed a drop in its genus *Akkermansia*, which has been described to be abundant in the gut of healthy subjects compared with obese patients or patients with metabolic syndrome. *Akkermansia* has also been identified as an important bacterium in the maintenance of gut mucosal integrity and defense against pathogenic bacteria [51]. It was shown to produce both propionate and acetate [52]. The significant age-dependent decrease in Akkermansia abundance in HFrD-fed mice might account for the reduction in SCFAs. In the same way, it might indicate an impaired gut health and instable mucus barrier, leading to increased translocation of bacteria and metabolites into blood circulation and, thus, supporting disease progression in HFrD-fed L2-IL1B mice. Furthermore, *Akkermansia* negatively correlated with dysplasia and positively with GC ratio and was enriched in CD-fed mice, highlighting that the shift in the microbiota was caused by the high-fructose diet and had an impact on the host.

The significant upregulation of pathways associated with the synthesis of nucleic acids, proteins, NAD(P) coenzymes, folates, and menaquinones in HFrD-fed mice in comparison with CD-fed mice reveal a fructose-dependent alteration of the microbial communities in these mice. Niacin is a B vitamin, known by its coenzymes NAD and NAD(P), NAD being essential for the synthesis and repair of DNA [53]. Folates are essential for vertebrates [54] and play a role in the biosynthesis of methionine, purine, and pyrimidine. Folate deficiency has been linked with multiple pathologies including cancer [55]. Together with folates, menaquinones (vitamin K_2_) are also essential for animals and are produced by bacteria in the intestine [56,57]. Furthermore, vitamin K2 has been shown to have inhibitory effects on cancerous cells [58,59]. Being the de novo biosynthesis of nucleic acids’ and proteins’ high energy-consuming mechanisms [60,61], the upregulation of these pathways might be the result of the excessive energy in the form of fructose. On the contrary, the upregulation of pathways involved in the biosynthesis of potentially protective vitamins might explain the mild effect of fructose on the acceleration of EAC progression. 

The observed pathway alteration within the microbial communities of HFrD-fed mice was mainly related to the biosynthesis of nucleic acids as well as to metabolic and anabolic pathways. While the activity of these pathways was higher in 3-month- and 9-month-old mice, it showed a relatively low activity in 6-month-old HFrD-fed mice. Despite the possibly related age-dependent shift in pathway activation to the accelerated dysplasia, reduced differentiation, and mucus production in the BE region of 6-month-old HFrD-fed mice, we cannot explain this contradictory trend. Considering that this observation resulted from an in silico prediction [62], further analysis evaluating the activity of actual microbial pathways more deeply are needed.

### 4.2. No Weight Gain Was Detected in HFrD Mice despite Increased Colonic Gluconeogenesis and Hepatic De Novo Lipogenesis

Regarding global effects attributed to dietary preferences, HFrD did not induce a significant weight gain in L2-IL1B mice, despite higher total food intake in these animals. A similar observation was made previously in the HFD study [8]. This finding was linked with evidence of increased energy turnover in HFrD-fed L2-IL1B mice, likely due to the systemic inflammation related to IL1B overexpression [8]. Furthermore, in a human study comparing diet-dependent weight gain between a glucose intervention group and a fructose intervention group, significant weight gain was observed in the glucose intervention group only [63]. This might also explain why the HFrD-fed mice of our study did not gain weight, as initially expected. Fructose overconsumption has been linked with prejudicial metabolic changes, mainly attributed to the role of fructose in de novo lipogenesis in the liver [40,64]. To assess the metabolic effects of a high fructose intake, the expression of key enzymes of fructose metabolism in the colon and liver was evaluated in this study. Considering the reactions catalyzed by the enzymes found to be overexpressed in theses tissues, our data suggest that fructose presumptively leads to increased colonic gluconeogenesis in 6-month-old HFrD mice and increased hepatic de novo lipogenesis in 9-month-old HFrD mice. Interestingly, this would also explain why the liver of 9-month-old HFrD-fed mice was significantly heavier than the liver of age-matched controls.

### 4.3. The Metabolic Profile of HFrD Mice Suggests a Diet-Related Alteration of Substrate Utilization by Host and Microbiome, as Well as a Limited Effect on Systemic Inflammation and Moderate Acceleration of the Dysplasia Phenotype

Untargeted metabolome analysis of feces, serum, and tissue revealed diet-specific changes in the metabolic profile of the mice. Deoxyhexoses were found in feces and serum of HFrD-fed mice. Deoxyhexoses are first metabolized into lactaldehyde, which depending on oxygen availability, can be reduced to 1,2-propanediol under anaerobic conditions or oxidized to lactate under aerobic conditions [65]. 1,2-propanediol can be metabolized through multiple steps into propionic acid [26]. Intensity levels of deoxyhexoses in feces correlating with the concentration of propionic acid in the fecal samples of HFrD-fed mice suggest a bacteria-mediated anaerobic metabolization of the deoxyhexoses in the intestine of these mice [26,65]. On the contrary, intensity levels of deoxyhexoses in serum correlated with lactic acid concentrations in the serum of HFrD mice. Considering the metabolic fate of deoxyhexoses in an aerobic environment [65] and the role of blood as an oxygen carrier [66,67], it can be assumed that the deoxyhexoses in the blood were further metabolized into lactic acid. Ketodisaccharides were found in fecal samples of only CD-fed mice. Certain bacteria are known to convert disaccharides into ketodisaccharides [68]. Disaccharides, branched alpha-dextrins, and trisaccharides are degradation products of complex carbohydrates such as starch [69,70,71]. Compared with the HFrD that only included single sugars and no complex sugars, the CD contained 45.9% of starch. The presence of ketodisaccharides in feces, as well as the enrichment of trisaccharides in the cardia tissue of CD-fed mice, can, thus, be associated with the dietary starch in CD. Moreover, hexoses enriched in the cardia tissue of HFrD-fed mice presumably indicate phenotypic deterioration in correlation with increased cellular energy turnover in BE epithelial cells, as often observed during carcinogenesis [72,73]. On the whole, untargeted metabolomic analysis indicates diet-related alterations in the metabolic profile, partially resulting from the microbial shift.

### 4.4. Levels of Protectively Acting SCFA Are Decreased in HFrD

In targeted metabolomic analyses, we further examined SCFA levels in stool and serum of the mice. SCFAs are the main metabolites produced by bacterial anaerobic fermentation of unabsorbed carbohydrates, dietary fiber, and proteins in the colon [74,75,76]. SCFAs function as systemic energy substrates with distinct metabolic roles [75]. Acetic- and propionic acid can be absorbed from the colon and processed in the liver. While acetate is used as a substrate for fatty acid synthesis [74], propionic acid has been shown to hinder this process [74,77], reduce immune system recruitment, and improve tissue insulin sensitivity [77]. Butyric acid is the preferred source of energy for colonocytes [78]. It has a positive impact on genetic/epigenetic modulations, signaling pathways, and immune response and has antineoplastic features, especially in colon carcinoma [75,78,79]. Isobutyric acid is generated by fermentation of branched amino acids from undigested proteins [80]. An increased colonic production of isobutyric and propionic acid has been associated with lowered glucose and insulin blood concentrations [81]. Valeric acid is generated from amino acids and has antimicrobial features. Furthermore, the oral administration of valeric acid protected against gastrointestinal inflammation and gut dysbiosis in animal models fed a high-fat diet or subjected to irradiation [82]. Compared with age-matched controls, we found that in HFrD-fed L2-IL1B mice, the concentration of these generally favorable SCFAs was reduced in fecal samples. Reduced levels of the different SCFAs in HFrD-fed mice was presumably the result of altered metabolic activity of the microbiota in HFrD mice. Butyric and lactic acid contents were also reduced in the serum of HFrD-fed compared with CD-fed mice. An overall reduction in SCFA indicates perturbation of the host metabolism by HFrD, as also found by gene expression analysis. The presence of SCFAs in the serum of the mice indicates a potential systemic effect of these metabolites by distribution via blood circulation, yet the role of SCFAs in disease progression in L2-IL1B mice is not yet clear. As SCFAs are microbially metabolized compounds, these findings correlate with our observations of a changed gut microbiome in HFrD-fed compared with CD-fed mice.

### 4.5. HFrD-Fed Mice Present a Moderate Inflammatory Phenotype in Comparison with HFD-Fed Mice

Evaluating the immune phenotype of HFrD mice, we found an age-dependent increase in pro-tumorigenic neutrophils and γδ T cells in the cardia of HFrD-fed mice, with comparatively lower numbers of neutrophils in HFrD-fed compared with CD-fed mice. Considering that γδ T cells have been shown to act defensively against pathogens and tumorigenesis [83,84], this observation together with the finding of metabolites with anti-inflammatory potential in HFrD mice explains the limited effect of the diet on systemic inflammation. As inflammation was found to be a crucial factor for disease progression in BE, especially in the L2-IL1B model, this finding might explain the modest acceleration of the dysplasia phenotype in HFrD-fed mice compared with HFD-fed mice. In the HFD study, the inflammatory phenotype was massively enhanced by the diet, thus explaining pronounced acceleration of the phenotype in HFD mice [8].

### 4.6. Tissue Remodeling, Metabolism, and Gut Barrier Protection Gene Sets Are Enriched in HFrD Mice While Pro-Oncogenic Gene Sets Are Enriched in CD Mice

To gain a better and deeper insight into molecular factors responsible for phenotypic changes between CD- and HFrD-fed mice, the gene expression profile in the SCJ was assessed. Significantly increased expression of *Aldolase B* suggests significantly increased fructose uptake of cells in the SCJ of HFrD-fed mice compared with controls. Enhanced energy metabolism is one of the hallmarks of cancer and, thus, correlates with promotion of the phenotype in HFrD mice. (PI3K)/AKT mTOR-, mTOR C1-signaling, and p53 signaling, which are involved in cell growth, survival, and replication, are often deregulated during carcinogenesis and were upregulated in CD-fed mice. [85,86,87,88]. In HFrD-fed mice, genes involved in EMT, angiogenesis, and hedgehog signaling but also in ECM remodeling were enriched, indicating structural changes in the tissue microenvironment promoting malignant disease progression in correlation with increased dysplasia scores and stem cell numbers [89,90,91]. Hedgehog signaling is a key biological process involved in embryonic development, stem cell maintenance, cell differentiation, and proliferation [92]. Thus, the enrichment of genes involved in hedgehog signaling might explain the increased stem cell levels in the BE epithelium of HFrD-fed mice. Enrichment of genes involved in metabolic diseases and bile acid metabolism indicate promotion of a metabolic phenotype and increased bile acid production induced by diet-dependent enhanced de novo lipogenesis, with bile acids being associated with promotion of disease progression in BE and EAC [93]. The gene expression profile of HFrD mice indicates higher metabolic activity and energy turnover in mice due to high sugar consumption. Enrichment of IgA production and O-linked glycosylation-associated pathways are presumptively protective reactions to changes in microbiome and nutrient availability through HFrD intake. Both pathways are important for tissue homeostasis. IgA is involved in non-inflammatory mucosal protection against microbial commensals, and O-linked glycosylation is a metabolic process involved in production of mucins, which protect the epithelial gut barrier, potentially due to the increased availability of sugar molecules [94,95]. Enrichment of these pathways indicates increased epithelial protection against microbiota and pathogens and might explain why the phenotype of HFrD-fed mice was not as pronounced as that seen in HFD-fed mice. 

## 5. Conclusions

A modern, westernized lifestyle includes a diet rich in both fat and sugars [96], which along with obesity, coincide with the increase in prevalence of noncommunicable diseases (NCDs) [97,98]. Although there are studies that have assessed esophageal microbiota and its changes along EAC progression [99,100,101], little is known about the link between EAC progression and intestinal microbiota. Dietary factors are critical determinants of gut microbiota diversity and plasticity [102]. Fat intake has long been the focus of research. However, research on the consequences of chronically high sugar consumption has recently been gaining more attention [103,104]. Saccharose and high-fructose corn syrup (HFCS) are the most commonly used types of refined sugar in the food industry [104]. Modelling the effects of fructose overconsumption on the BE-mouse model, we found that a HFrD accelerated the phenotypic progression of BE to dysplasia in correlation with reduced differentiation and enhanced Lgr5 stem cell expansion, especially in younger animals. However, these effects were less pronounced than those seen in HFD-fed IL-1B [8]. We also found a limited impact of HFrD feeding on the inflammatory phenotype in comparison with CD-fed mice. Similar to findings in HFD-fed mice, in HFrD-fed mice, we found effects on the gut, serum, and tissue metabolome, as well as diet-induced alterations of the composition, functionality, and metabolism of intestinal microbial communities. Our findings underline the impact of dietary patterns on the health and balance of the metaorganism. The outcomes from this study in addition to the HFD study [8] provide important insights into the ways in which diet might influence disease progression and highlight the effect of diet on the intestinal microbiota profile and balance in the BE-mouse model. However, studying dietary patterns rather than individual dietary components would better capture the complex effects of dietary habits [105] on EAC. A further limitation of this study is the use of a mouse model. Despite the importance of mice in biomedical research, the bench-to-bed translational power of mouse studies is arguable, since mice differ from humans in many aspects. Furthermore, we only looked for diet-related changes in the intestinal microbiota and not in the esophageal microbiota itself. Although the effect of targeting the intestinal microbiota or its metabolism as prophylactic or therapeutic approaches for EAC is unclear, it is evident that diet has an impact on the overall health of the host and the balance of its microbiome. In addition to gaining a better understanding of the effects of diet on noncommunicable diseases such as EAC, it is necessary to actively encourage people to strive for a healthy, non-sedentary lifestyles, maintain a balanced diet, and, thus, reduce the risk of NCDs.

## Figures and Tables

**Figure 1 microorganisms-09-02432-f001:**
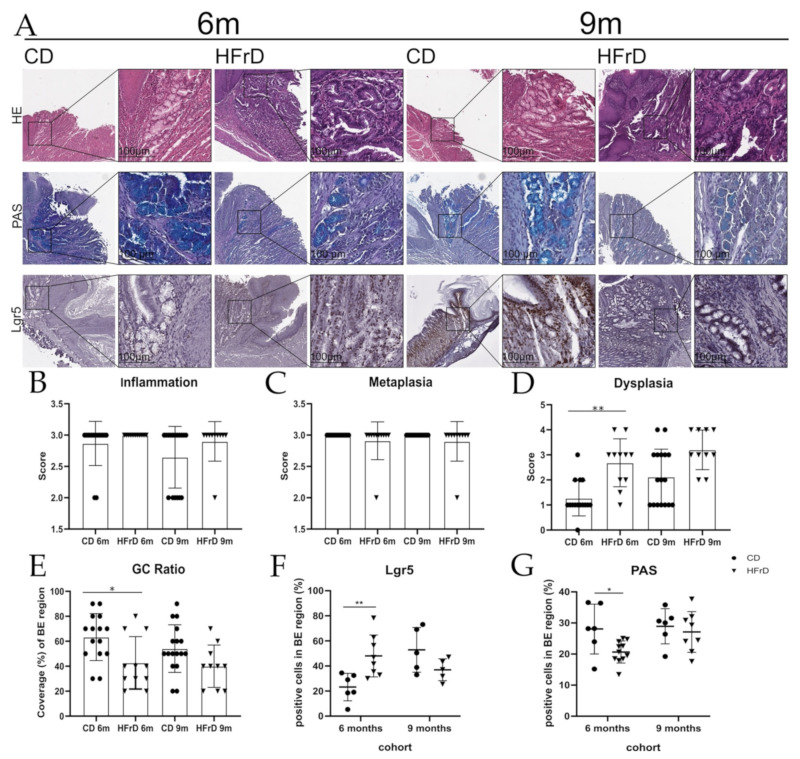
Histologic tumor scores, stem cell expansion, and proportion of mucus-producing goblet-like cells in the BE region. (**A**) Microscopic pictures of HE, PAS, and Lgr5 staining; analysis of histologic tumor scores of the BE region: (**B**) inflammation score, (**C**) metaplasia score, (**D**) dysplasia score, and (**E**) goblet-cell (GC) ratio. Inflammation and metaplasia-scores of mice from the different intervention groups and cohorts did not differ. While the dysplasia scores of 6-month-old HFrD mice significantly increased (*p* = 0.0079) in comparison with CD mice, the GC ratio was significantly smaller (*p* = 0.0435) in 6-month-old HFrD compared with CD mice. In 9-month-old mice, dysplasia scores and GC ratio failed to reach significance; (**F**) analysis of Lgr5-positive cells in the BE region. While the percentage of Lgr5-positive cells was significantly higher (*p* = 0.0083) in 6-month-old HFrD-fed mice than in CD mice, there was no difference in the older mice; (**G**) analysis of mucus-producing goblet-like cells in the BE region. While the area of PAS-positive cells in the BE region was significantly smaller in 6-month-old HFrD compared with CD mice (*p* = 0.0176), the effect did not reach significance in the 9-month-old cohort. (**B**–**E**): Statistical comparison between intervention groups and age cohorts. Data are represented as mean ± SD. For the statistical analysis of inflammation, metaplasia, and dysplasia scores, a Kruskal–Wallis test with Dunn’s multiple comparisons test was performed, and for the statistical analysis of the GC ratio, an ordinary one-way ANOVA with Holm–Sidak’s multiple comparisons test was performed. (**F**–**G**): Statistical comparison between intervention groups and age cohorts. Data are represented as mean ± SD. For the statistical analysis, an unpaired *t*-test was used. (**B**–**G**): circles represent samples from CD-fed mice and triangles represent samples from HFrD-fed mice. Significance level: * (significant); ** (very significant).

**Figure 2 microorganisms-09-02432-f002:**
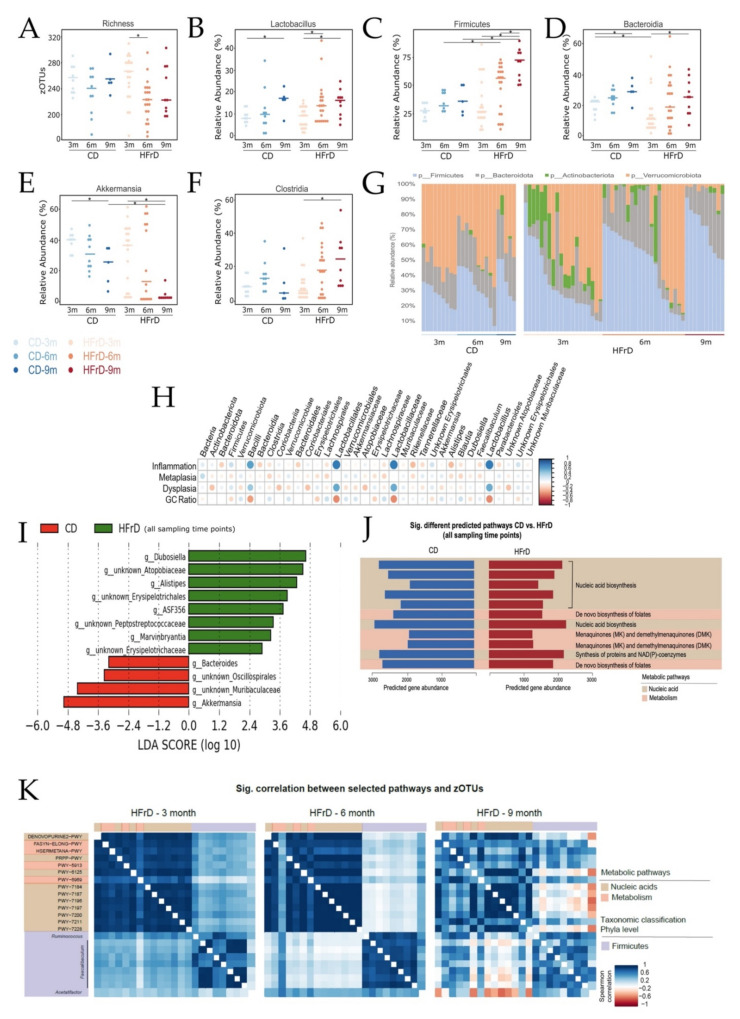
Graphical representation of richness and abundance of Lactobacillus, Firmicutes, Bacteroidia, Akkermansia, and Clostridia in gut microbiome. (**A**) Richness. HFrD-fed mice showed an age-dependent decrease in richness (pHFrD:3–6m = 0.0024); (**B**) relative abundance (%) of Lactobacillus. In both intervention groups, Lactobacillus showed an age-dependent increase (pCD:3–9m = 0.0290, pHFrD:3–9m = 0.0053; pHFrD:3–6m = 0.0091); (**C**) relative abundance (%) of Firmicutes. HFrD-fed mice showed an age-dependent increase in Firmicutes (pHFrD:3–9m = 0.0003 and pHFrD:6–9m = 0.0132). Additionally, the relative abundance of Firmicutes increased significantly in HFrD-fed mice of both cohorts, compared with age-matched controls (pCD6m–HFrD6m = 0.0433 and pCD9m–HFrD9m = 0.0027); (**D**) relative abundance (%) of Bacteroidia. In both intervention groups, an age-dependent increase was observed (pCD:3m-9m = 0.0190 and pHFrD:3m-9m = 0.0083). Additionally, the relative abundance of Bacteroidia decreased significantly in HFrD-fed mice in comparison with CD-fed mice (*p* = 0.0201); (**E**) relative abundance of Akkermansia. In both intervention groups, an age-dependent decrease was observed (pCD:3–9m = 0.0120 and pHFrD:3–9m = 0.0061). In addition, the relative abundance of Akkermansia decreased significantly in 9-month-old HFrD-fed mice compared with controls (pCD9m–HFrD9m = 0.0040); (**F**) relative abundance of Clostridia. In HFrD-fed mice, the relative abundance of Clostridia increased in an age-dependent-manner (pHFrD:3–9m = 0.0016); For the statistical analysis, a pairwise Wilcoxon rank sum test was performed; (**G**) graphical representation of phyla distribution in CD-fed mice (left) and in HFrD-fed mice over time (right). While there is an overall increase in Firmicutes and Bacteroidota, the ratio of Verrucomicrobiota decreases in both intervention groups over time; (**H**) bacteria correlating with histologic tumor scores and goblet-like cell (GC) ratio. Bacilli, Lactobacillales, Lactobacillaceae, and Lactobacillus correlated positively with inflammation and dysplasia and negatively with GC ratio; (**I**) bar plot of the LDA scores resulted from the linear discriminant analysis effect size (LEfSe) of genera differentially abundant between CD- and HFrD-fed mice (all sampling timepoints included); (**J**) prediction of significantly different activated pathways in intestinal microbial communities of CD and HFrD mice. Pathways associated with the synthesis of nucleic acids, proteins, NAD(P) coenzymes, folates (vitamin B9), and menaquinones were found in HFrD-fed mice in comparison with CD-fed mice (all sampling timepoints included); (**K**) heatmaps representing age-related functional and metabolic differences in intestinal microbial communities of HFrD-fed mice. Pathways found to be altered were the TCA pathway and pathways associated with the synthesis of nucleic acids, methionine, and synthesis and elongation of fatty acids. Significance level: * (significant).

**Figure 3 microorganisms-09-02432-f003:**
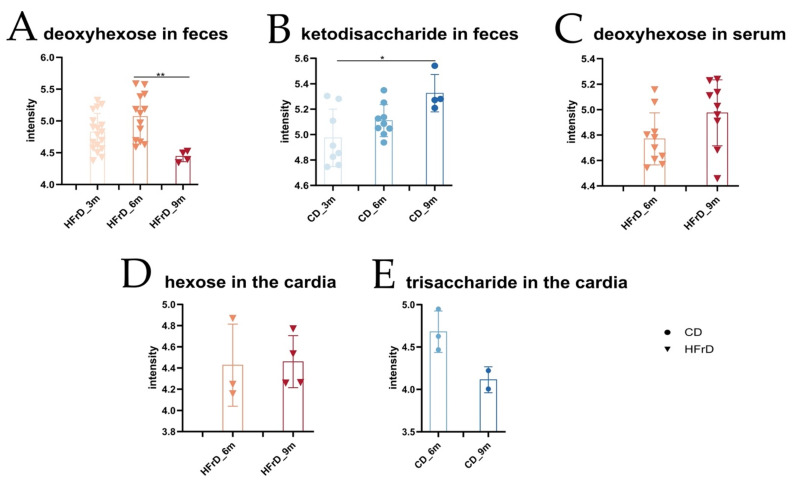
Untargeted metabolomic profiles in feces, serum, and cardia tissue. Considered metabolites were found in either the CD-fed or the HFrD-fed mice. (**A**) Deoxyhexose in fecal samples of HFrD-fed mice. Content of deoxyhexose decreased significantly between samples of 6-month-old and 9-month-old mice (*p* = 0.0032); (**B**) ketodisaccharide in fecal samples of CD-fed mice. Content of ketodisaccharide increased significantly between samples of 3-month-old and 9-month-old mice (*p* = 0.0110); (**C**) deoxyhexose in serum samples of HFrD-fed mice. Content of deoxyhexose did not differ significantly between the age groups; (**D**) hexose in cardia tissue of HFrD-fed mice. Content of these monosaccharides did not differ significantly between the age groups; (**E**) trisaccharide in cardia tissue of CD-fed mice. Content of trisaccharide did not differ significantly between the age groups; statistical comparison between the age cohorts. Data are represented as mean ± SD. (**A**,**B**): Depending on the normal distribution of the data, either an ordinary 1-way ANOVA with Tukey’s multiple comparisons test or a Kruskal–Wallis Dunn’s multiple comparisons test was performed for the statistical analysis. (**C**–**E**): Depending on the normal distribution of the data, either an unpaired *t*-test or a Mann–Whitney test was used for the statistical analysis. (**A**–**E**): circles represent samples from CD-fed mice and triangles represent samples from HFrD-fed mice. Increasing blue shades correspond to CD-fed mice (3m to 9m) and increasing red shades correspond to HFrD-fed mice (3m to 9m). Significance level: * (significant); ** (very significant).

**Figure 4 microorganisms-09-02432-f004:**
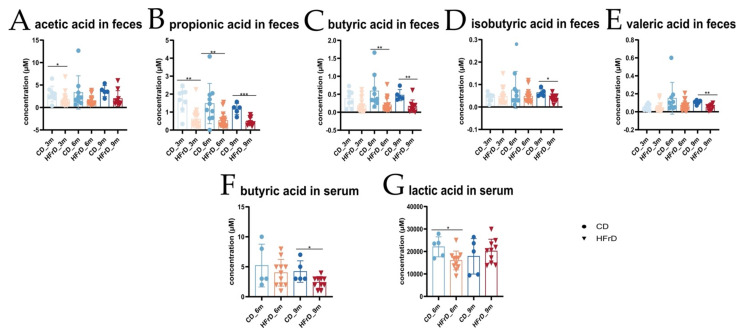
SCFA profiles in fecal samples taken at different timepoints and serum samples taken at dissection day. (**A**) Acetic acid in feces. Content decreased significantly in samples of 3-month-old HFrD-fed mice in comparison with CD-fed mice (*p* = 0.0278); (**B**) propionic acid in feces. Compared with age-matched controls, content decreased significantly in 3-month- (*p* = 0.0017), 6-month- (*p* = 0.0022), and 9-month-old HFrD-fed mice (*p* = 0.0003); (**C**) butyric acid in feces. Content decreased significantly in 6-month- (*p* = 0.0020) and 9-month-old HFrD-fed mice (*p* = 0.0080) compared with controls; (**D**) isobutyric acid in feces. Content decreased in HFrD-fed mice of all age stages in comparison with age-matched controls. However, the difference was only significant in the 9-month-old cohort (*p* = 0.0195); (**E**) valeric acid in feces. In all age stages, content tended to be lower in HFrD-fed mice than in CD-fed mice. Here, the difference was also significant in only the 9-month-old cohort (*p* = 0.005); (**F**) butyric acid in serum. Content was significantly lower in 9-month-old HFrD-fed mice than in 9-month-old CD-fed mice (*p* = 0.0231); (**G**) lactic acid in serum. Content decreased significantly in 6-month-old HFrD-fed mice in comparison with 6-month-old CD-fed mice; statistical comparison between intervention groups and age cohorts. Data are represented as mean ± SD. Depending on the normal distribution of the data, either an unpaired *t*-test or a Mann–Whitney test was used for the statistical analysis. (**A**–**G**): circles represent samples from CD-fed mice and triangles represent samples from HFrD-fed mice. Increasing blue shades correspond to CD-fed mice (3m to 9m) and increasing red shades correspond to HFrD-fed mice (3m to 9m). Significance level: * (significant); ** (very significant); *** (extremely significant).

**Figure 5 microorganisms-09-02432-f005:**
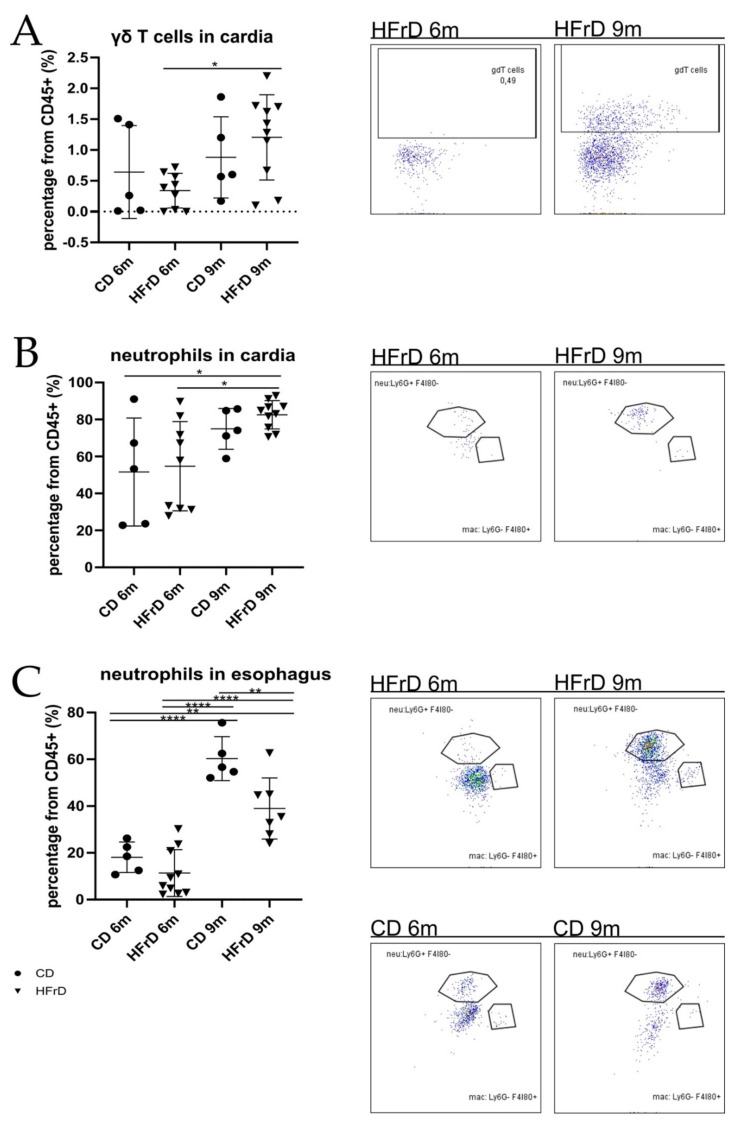
Myeloid and T cell populations in gastroesophageal tissues. (**A**) γδ T cell infiltration in cardia tissue. While there were no differences between the intervention groups of both age groups, the γδ T cell infiltration in HFrD-fed mice was significantly higher in 9-month-old than in 6-month-old mice (*p* = 0.0250); (**B**) neutrophil infiltration in cardia tissue. While there were no differences between the intervention groups of both age groups, the infiltration of neutrophils in HFrD-fed mice was significantly higher in 9-month-old than in 6-month-old mice (*p* = 0.0235); (**C**) neutrophil infiltration in esophageal tissue. In both intervention groups, the infiltration of neutrophils increased significantly (*p*_CD|HFrD_ < 0.0001) in an age-dependent manner. In general, the neutrophil infiltration in HFrD-fed mice of both age groups tended to be lower than in their respective age-matched controls. This difference was, however, significant in the 9-month-old cohort (*p* = 0.0054); statistical comparison between the age cohorts. Data are represented as mean ± SD. Depending on the normal distribution of the data, either an ordinary 1-way ANOVA with Holm–Sidak’s multiple comparisons test or a Kruskal–Wallis with Dunn’s multiple comparisons test were performed for the statistical analysis. (**A**–**C**): circles represent samples from CD-fed mice and tiangles represent samples from HFrD-fed mice. Significance level: * (significant); ** (very significant); **** (extremely significant).

**Figure 6 microorganisms-09-02432-f006:**
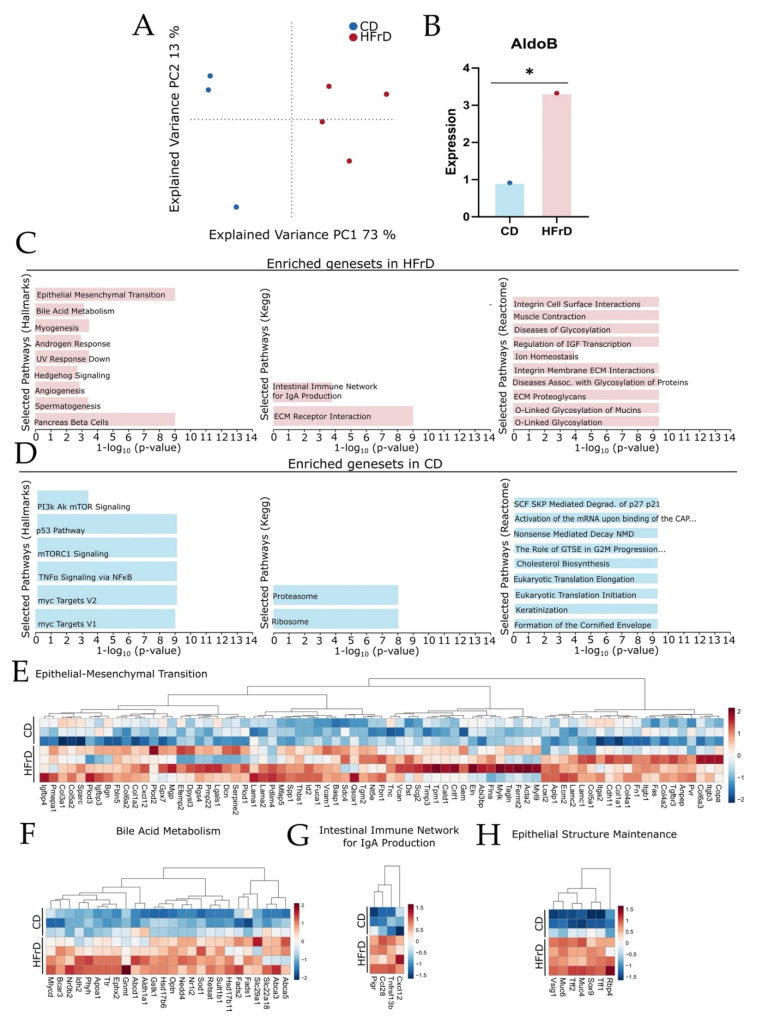
Gene expression in BE epithelium is significantly altered between CD and HFrD mice. The gene expression profile of 9-month-old CD (*n* = 3) and HFrD (*n* = 4) mice was analyzed by RNA-sequencing. (**A**) Principal component analysis (PCA) plot of gene expression profiles between CD and HFrD-fed mice; (**B**) mean gene expression levels of Aldolase B in CD and HFrD mice depicted as red and blue circles. Expression is significantly altered between CD and HFrD mice (*p*_adj_ = 0.0105); (**C**) top 10 significantly enriched MsigDB Hallmark, Kegg DB, and Reactome DB gene sets identified by gene set enrichment analysis (GSEA) in HFrD treatment; (**D**) top10 significantly enriched MsigDB Hallmark, Kegg DB, and Reactome DB gene sets identified by gene set enrichment analysis (GSEA) in CD treatment; (**E**–**H**): significantly enriched gene sets in HFrD mice depicted as heatmaps showing gene expression changes among samples. Statistical details about the gene set pathways are provided in the supplements. Significance level: * (significant).

## Data Availability

Raw data resulting from 16S rRNA gene sequencing were deposited to the NCBI Sequence Read Archive [http://www.ncbi.nlm.nih.gov/bioproject/769229; accessed on 7 October 2021] under the accession no. PRJNA769229. Raw data resulting from RNA-sequencing were deposited to the ENA Archive [https://www.ebi.ac.uk/ena/browser/view/PRJEB48315; accessed on 5 November 2021] under the accession no. PRJEB48315.

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
