# Peer review of "High-Fructose Diet Alters Intestinal Microbial Profile and Correlates with Early Tumorigenesis in a Mouse Model of Barrett’s Esophagus"

_microorganisms, 2021, doi:10.3390/microorganisms9122432_

Round 1

Reviewer 1 Report

The authors have corrected all flaws of the manuscript thoroughly and consistently. Now the article seems much more clear and improved. But I have few small recommendations.

1) Lines 133-153. The authors freely use and mix different terms: zOTUs, and ASVs. But in fact, all these things, plus OTUs, are produced by different tools. For instance, UPARSE used by the authors, is a tool producing only OTUs, not zOTUs  or ASVs. Please, use only term OTUs along the text of the manuscript.

2) Lines 139-140. "For phylogenetic analyses, maximum-likelihood trees
were generated by FastTree based on MUSCLE alignments in MegaX [15]." If the authors do not give and discuss phylogenetic trees in the manuscript, they should remove this sentence.

3) Line 206. "Appendix A: Supplementary Methods" is mentioned here, but it is absent in the Supplementary. Remove the sentence if all methods are well described in the main body of the article.

4) Both attachments "Supplementary file(s)" and "Non-published material" contain the same files. Besides, in both archives there are Excel files with an OTU table "Proano-Vasco-2021_Resubmission microorganisms-1341930_zOTUs_Table-norm_as supplement", which are failed to be open. Please, put in the Supplementary the Excel file, which is available for opening.  

5) Fig. 2I Designations of bars and both axis are too small and not visible.

6) Lines 279-288. For analysis of functional profiling with PICRUST the authors should use relative abundance, but not abundance. Abundance reflects an absolute number of genes and depends on coverage depth, which varies in wide ranges between the samples. As exception, abundance may be used after previous normalization of read numbers in all samples by a minimal read number. In such case it should be directly indicated as well as level of normalization in the section 2.3. After that, the authors should reanalyse the data on functional profiling.

7) Line 509. Exchange "genera Akkermansia" by "genus Akkermansia".

Reviewer 2 Report

The manuscript improved overall after revision. There are a few minor points that may require attention:

  • Legends missing to Figure 1F and 1G
  • It is still not clear why the authors are presenting selected bacteria in Fig. 2-F. It is the results from a statistical testing or selected results and if so what are the selection criteria to present these data?
  • Labels of Fig 2G  and K are not visible
  • Fig 2I: is LefSe done on all time points or selected time points and what was the rational for selecting specific time points?
  • Fig. 5 and 6 are blurry and cannot be read.
  • Title needs to be revised and toned down as a) HFrD does not accelerate tumorigenesis and b) not at all through changes in the microbiome - at least there is no mechanistic evidence for this claim in the manuscript; only correlation.

Author Response

This manuscript is a resubmission of an earlier submission. The following is a list of the peer review reports and author responses from that submission.

Round 1

Reviewer 1 Report

In the present manuscript, the authors investigate the effects of long-term high-fructose diet (HFrD) on tumor development and associated pro-carcinogenic mucosal changes as well as fecal microbiome changes in an animal model of Barrett’s esophagus (IL2-IL1B mice). HFrD is considered a major component of westernized diet that has been linked to obesity which, in turn, favors carcinogenesis, but the effects of fructose itself are not well studied in cancer.

In the present study, the effects of fructose-rich diet are overall rather modest in the tumor model studied. In addition to increased dysplasia scores, an expansion of stem cells at the SC junction, changes in the expression of several genes, in serum and fecal metabolites and in a few taxa of the gut microbiome.  

The authors compiled a considerable amount of work, and all these efforts deserve appreciation. However, the manuscript is currently a piece of loosely linked fragments rather than “a story” and excessively descriptive. For instance, it is not entirely clear whether the authors put their focus on the diet, or on the microbiome changes or age-related changes that are also heavily present in the results.

Major points that would deserve consideration:

  • As far as I understand, the authors did not include any wildtype littermate mice, but used only IL2-IL-1B mice, which makes any assessment of diet effects on the host (at least at state) difficult.
  • The microbiome analysis deserves a deeper level of characterization including alpha diversity (with Shannon or Simpson inverse), beta-diversity analyses (PCoAs etc) including PERMANOVAs for statistical testing, and most important, an unbiased analysis of taxonomy as it is not clear whether the authors picked these taxa or whether they were revealed by a pairwise, cross-cohort / time point analysis with FDR corrections. Along these lines, were the correlation analyses correct for multiple testing?
  • As the authors measure gut taxonomy and serum / fecal metabolites, it would be very interesting to mine the gut microbiome for gene pathways encoding metabolic pathways which would explain the presence / levels of metabolites observed. Even if shotgun metagenome analysis were not applied, one could at least run a Picrust analysis.
  • Figure 3C also deserves revision as the control values are missing in almost every figure or were these metabolites not detected in various groups?
  • The RNA seq showed very interesting results, and could be a starting point for any more mechanistic studies.

Reviewer 2 Report

The manuscript ‘High-Fructose Diet accelerates early tumorigenesis in a Mouse Model of Barrett´s Esophagus’ contains original and significant scientific data. However, in its current form it partially meets the aim and scopes of the ‘Microorganisms’ journal. This manuscript reports on changes of multiple characteristics in the BE-mouse model accompanied with a high fructose diet (HFrD) including histologic features, inflammatory status, metabolism, and composition of gut microbiota. At the same time the manuscript in whole seems as an experimental cancer research focused on the histologic and metabolic aspects of tumorigenesis added by a minor fragment of gut microbiome analysis. Though this data is of a great interest for oncologists, a week emphasis on the microbiological findings alone the manuscript is a basic issue preventing its publication in current form in this journal. Moreover, I do not think that the manuscript in its current form would be attractive for a wide range of the journal readers and for microbiologists generally. So, I strongly recommend a major revision of the manuscript.

The following are some comments and suggestions that may improve the manuscript and supplementary data.

1. The Title in current form is not in aim and scopes of the ‘Microorganisms’ journal at all. So, it needs to be corrected through including of an aspect related to the gut microbiota of model mice.

2. The abstract does not reflect fully the main findings. The authors have to include in the first part of the abstract (lines 22-29) some known facts on diet-related microbial factors of tumorigenesis in digestive tract. At the same time, the second part of the abstract (lines 29-37) focused on the main results, should be expanded with the most significant changes revealed in the gut microbiota and metabolism.

3. The abstract contains a conclusion, which is not supported ‘recent evidence linking the intake of refined sugar with disease’ (lines 27-28). This conclusion should be accompanied by detailed description in Introduction section and references, accordingly. Please, clarify, which ‘disease’ did the authors mean in this case?

4. Introduction contains a detailed description of the esophageal adenocarcinoma (EAC) risk factors (lines 49-61) and a short description of a recent study conducted by the authors and devoted to the effects of a high-fat diet on the progression of Barret’s Esophagus in the L2-IL1B mouse model (lines 62-73). Also the role of glucose in cancer metabolism and tumor proliferation is mentioned fluently (lines 75-76). Unfortunately, this background does not seem to be significant and enough for substantiation of the experimental design used in the reviewed study. To provide more justification the authors have to search thoroughly any possible relations between HFrD and development or progression of EAC or another gastrointestinal cancer. Another set of facts that should be revealed and listed by the authors in Introduction section concerns the role of gut microflora, established or supposed, in development or progression of EAC or another gastrointestinal cancer.

5. Material and methods (M&M) section contains some unclarities, which prevent to consider the study reproducible. First of all, how many animals were in every group and every cohort (lines 89-92)? Were the samples taken from every animal for microscopy, immunohistochemical tests, metagenomics, metatranscriptomics, metabolomics, etc. If not, precise numbers of samples for every test and every cohort should be indicated.

6. Raw reads obtained after 16S rRNA gene sequencing should be deposited in SRA archive of NCBI or another open depository of DNA sequences together with metadata for every sample sequenced. Links and accession numbers should be indicated for the deposited data in the M&M section (line 117).

7. Description of 16S rRNA gene sequencing (lines 109-119, 621-652) is not sufficient. Some details characterizing quantity and quality of the sequencing data should be indicated in the main body of the article or in supplementary such as number of raw reads for every sample and its variation, number of reads for analysis for every sample and its variation, number of chimeric sequences. Were the contaminating reads detected, and by which way? What database was used for taxonomic assignment of zOTUs?

8. Rarefaction curves for every sample should be provided in supplementary, to give reliable proof that differences between the groups by alfa diversity are not related with differences between the samples in their sequencing depth.

9. Description of fecal samples collecting for 16S rRNA gene sequencing should be given in details to convince a reader that the results were not affected by issues of sampling and storage. Thus, the authors have to indicate a period of time or its ranges between euthanasia and fecal sampling, used tools for sampling, and a part of intestine studied. Were the stool samples stored before DNA isolation? What was time and conditions of the storage? Were negative and positive controls used for sampling and consequent 16S rRNA sequencing? If yes, they also need to be indicated in M&M section or in supplementary.

10. In line 621 exchange ’16. S-rRNA-gene amplicon sequencing’ by ‘High-throughput sequencing of 16S rRNA gene amplicons’.

11. In line 626 indicate name of a shaker used.

12. In line 640 exchange ’16s’ by ‘16S rRNA gene’.

13. In line 641 exchange ’16sRNA’ by ‘16S rRNA gene’.

14. In line 660 exchange ’10.000’ by ’10,000’.

15. Results section contains comprehensive description of the main results and accompanied by necessary figures, tables and supplementary. Some recommendations you can find below. First of all, description of the gut microbiota in lines 186-229 is too short. I recommend to give a general view of the main bacterial taxa in control and in HFrD mice, e.g. as pie or bar charts. Also, an Excel table of zOTUs with size and taxonomy assigned will be useful as supplementary. Also, additional characteristics of gut microbiota diversity with Chao1, Shannon, Simpson indices, as well as description of dominant taxa would be useful.

16. What are units of richness (line 189, Fig. 2A), zOTUs or species? How did the authors obtain ‘normalized richness’ (Fig. 2A)? Please, correct the following parts of the manuscript.

17. Contradiction between Figs. 2C and 2F needs additional explanation. Fig. 2F shows increase in relative abundance of Clostridia between 9m-CD and 9m-HFrD cohorts by about 20%, in contrast to difference in relative abundance of Firmicutes between the same cohorts by about 35-40% (lines 196-198). The authors try to explain it in Discussion section (lines 401-403), but unsatisfactory. To clarify this issue, the authors have to mention other representatives of Firmicutes demonstrating a trend of relative abundance between the cohorts that is similar to Clostridia.

18. The following contradiction should be addressed. Figs. 2E and 2F show absolutely contrast trends in relative abundance between Akkermansia and Clostridia. Nevertheless, surprisingly, they both correlated negatively with dysplasia scores and positively with GC ratio (lines 206-208).

19. In Discussion section all results are interpreted and discussed thoroughly. However, it may be improved. Particularly, the comparison of the results obtained in this study with known data about role of microbiota in development and progression of EAC and other types of gastrointestinal cancer would be useful. Next question should be addressed. What are limitations of this study and its possible significance for prophylaxys, diagnostics and treatment of EAC in humans? May microbiota and metabolic pathways involving microbiota serve as targets for choice of diagnostic and treatment tools at EAC using the L2-IL1B mouse model?

20. What do % mean in lines 397? Do they reflect proportion of species or relative abundance? The authors should clarify this aspect.